# The Role of Lymph-Adipose Crosstalk in Alcohol-Induced Perilymphatic Adipose Tissue Dysfunction

**DOI:** 10.3390/ijms251910811

**Published:** 2024-10-08

**Authors:** Kourtney D. Weaver, Liz Simon, Patricia E. Molina, Flavia Souza-Smith

**Affiliations:** Department of Physiology, Louisiana State University Health Sciences Center-New Orleans, New Orleans, LA 70112, USA; kweav1@lsuhsc.edu (K.D.W.); lsimo2@lsuhsc.edu (L.S.); pmolin@lsuhsc.edu (P.E.M.)

**Keywords:** adipose, lymphatics, lymph, alcohol, inflammation, adipokines

## Abstract

Chronic alcohol use leads to metabolic dysfunction in adipose tissue. The underlying mechanisms and the contribution of alcohol-induced adipose tissue dysfunction to systemic metabolic dysregulation are not well understood. In our previous studies, we found that chronic alcohol feeding induces mesenteric lymphatic leakage, perilymphatic adipose tissue (PLAT) inflammation, and local insulin resistance in rats. The goal of this study was to further explore the link between alcohol-induced lymphatic leakage and PLAT immunometabolic dysregulation, locally and systemically, using in vivo and ex vivo approaches. Male rats received a Lieber–DeCarli liquid diet, of which 36% of the calories were from alcohol, for 10 weeks. Time-matched control animals were pair-fed. Adipokine levels were measured in PLAT, subcutaneous fat, plasma, and mesenteric lymph samples. Glucose tolerance was assessed after 10 weeks. Further, we used a novel ex vivo lymph-stimulated naïve PLAT explant approach to modeling lymph leakage to assess changes in adipokine secretion and expression of proinflammatory markers after stimulation with lymph from alcohol- or pair-fed animals. Our data show that chronic alcohol-fed rats presented PLAT-specific decreases in adiponectin and leptin levels, alterations in the expression of genes involved in lipid metabolic pathways, and associated impaired whole-body glucose homeostasis. Further, we found that direct naïve PLAT stimulation with lymph contents from alcohol-fed animals increased IL-6 expression in demonstrating the ability of lymph contents to differentially impact naïve adipose tissue. Overall, chronic alcohol feeding leads to depot-specific alterations in metabolic profile, impaired systemic glucose tolerance, and lymph-induced adipose tissue inflammation. The specific lymph components leading to PLAT immunometabolic dysregulation remain to be determined.

## 1. Introduction

Adipose tissue is an insulin-sensitive organ that plays an important role in the regulation of whole-body metabolism and glucose homeostasis. Adipose tissue stores energy as lipids and regulates their mobilization and distribution throughout the body [1]. Moreover, through the endocrine secretion of bioactive adipokines and cytokines, adipose tissue has the potential to modulate distant organs’ function and impact metabolic processes [2]. Chronic alcohol administration has been shown to alter adipose tissue structure and function in human, non-human primate, and rodent studies [1,3,4]. The mechanisms driving alcohol-induced adipose tissue dysfunction and whether this contributes to systemic metabolic dysregulation are not well understood and warrant further investigation.

Insulin resistance (IR) in adipose tissue often occurs before it develops in other insulin-sensitive tissues such as skeletal muscle and liver [5]; however, the mechanisms of adipose tissue-induced systemic IR are not well defined. Adipokines, such as adiponectin and leptin, play a key role in regulating metabolism, and alterations in adipokine secretion have been linked to IR [6]. Both binge and chronic alcohol administration in rodent models decreases plasma adiponectin levels [7,8,9], while only chronic alcohol consumption decreases adiponectin in humans [10]. Additionally, alcohol has been shown to alter leptin levels [11,12]. Finally, alcohol promotes lipolysis primarily due to the lack of inhibitory action of insulin rather than the stimulatory actions of catecholamines releasing free fatty acids (FFAs) from adipose tissue [13,14].

Adipose tissue depots differ in metabolic properties, immune profiles, and venous drainage [15]. Visceral fat, compared to subcutaneous fat (SFAT), is more metabolically active, has higher immune cell content, and is linked to systemic IR and inflammation [16]. Mesenteric adipose tissue (true visceral fat in rodents) is intimately associated with the mesenteric lymphatic system, draining into the portal vein [17], and delivering FFAs, cytokines, and adipokines directly into the liver. Collecting lymphatic vessels from the gut course through the mesenteric adipose tissue (perilymphatic adipose tissue-PLAT) serves as a conduit for dietary fats and nutrients absorbed from the gut and circulating immune cells [18].

Our previous studies demonstrated that exposure of rats to repeated alcohol binges as well as to a chronic alcohol diet leads to mesenteric lymphatic vessel hyperpermeability and subsequent leakage of lymph contents into PLAT. This was associated with the immunometabolic changes in PLAT, characterized by increased proinflammatory cytokine expression, immune cell infiltration, insulin signaling impairment, and decreased glucose uptake. Importantly, none of these alterations were observed in subcutaneous fat adipose tissue (SFAT), suggesting that PLAT immunometabolic dysregulation is likely a consequence of alcohol-induced lymphatic leakage [19,20]. Other models of mesenteric lymphatic leakage, including high-fat diet (HFD) feeding to mice, have demonstrated the ability of lymph contents to differentially impact adipose tissue metabolism in vivo and ex vivo [21], leading to insulin resistance. These findings and ours emphasize the importance of further investigating the direct effects of mesenteric lymph on mesenteric adipose tissue, as well as identifying circulating mediators in the lymph that have the potential to impact tissue physiology in the setting of mesenteric lymphatic leakage.

The goal of this study was to further explore the link between alcohol-induced lymphatic leakage and PLAT immunometabolic dysregulation. We tested the hypothesis that chronic alcohol feeding would alter adipokine expression in PLAT, but not SFAT, alter markers of PLAT lipid metabolism, and that these would be associated with impaired systemic glucose tolerance. Further, we hypothesized that ex vivo exposure to lymph contents from alcohol-fed animals would alter adipokine levels and increase inflammatory cytokine expression in naïve PLAT explants. Our data identified PLAT-specific alterations in adipokine levels and markers of PLAT lipid metabolism in chronic alcohol-fed rats and these were coupled with changes in whole-body glucose tolerance after alcohol feeding. Additionally, we found that chronic alcohol feeding induced changes in circulating lymph adiponectin. Using a novel lymph-stimulated naïve PLAT explant approach, we modeled lymph leakage ex vivo and found that direct stimulation with lymph contents from alcohol-fed animals increased proinflammatory cytokine expression in naïve PLAT. These findings provided additional support for the role of lymph-adipose crosstalk in alcohol-mediated adipose tissue dysfunction.

## 2. Results

### 2.1. Lymph Alcohol Concentrations Correlate with Blood Alcohol Concentrations

Alcohol concentration was assessed after 10 weeks of alcohol feeding in both blood and mesenteric lymph. Blood alcohol concentrations averaged 19.51 mM ± 2.48 in alcohol-fed animals (Figure 1A). Similarly, lymph alcohol concentration averaged 23 mM ± 3.35 in alcohol-fed animals (Figure 1B) and correlated to blood alcohol concentration in both groups with a Pearson correlation coefficient r = 0.9785, *p* ≤ 0.0001 (Figure 1C).

### 2.2. Chronic Alcohol Feeding Impairs Systemic Glucose Tolerance

To assess the effects of chronic alcohol feeding on systemic glucose tolerance, an intraperitoneal (IP) glucose tolerance test was performed. Prior to IP glucose injection, fasting blood glucose levels averaged 77 mg/dL ± 3.10 in pair-fed and 71.5 mg/dL ± 3.85 in alcohol-fed animals. Glucose administration resulted in a significant increase in blood glucose levels (189.5 mg/dL ± 39.35) in the pair-fed animals. This response was enhanced in alcohol-fed animals, reaching a peak of 262.33 mg/dL ± 21.26. Two-way ANOVA showed a main effect of alcohol (*p* < 0.005) (Figure 2A). The total area under the curve (AUC) calculated from the glucose concentrations measured following IP glucose injection was significantly higher in alcohol-fed animals compared to control animals; the average AUC for alcohol-fed animals (n = 6) was 26.8% higher than the AUC for control animals (n = 4) (Figure 2B).

### 2.3. Chronic Alcohol Feeding Dysregulates Adipokine Levels

To examine the impact of chronic alcohol on adipokine levels, we measured adiponectin and leptin in PLAT, SFAT, plasma, and mesenteric lymph. Adiponectin levels in PLAT from alcohol-fed animals were significantly lower compared to pair-fed controls (*p* = 0.0012) (Figure 3A). Adiponectin levels in SFAT (Figure 3B) were not significantly different between alcohol- and pair-fed animals (*p* = 0.1196). Adiponectin levels in mesenteric lymph samples (Figure 3D) were significantly lower in alcohol-fed animals (*p* = 0.0012). However, blood levels of adiponectin were not significantly different between groups (*p* = 0.0989) (Figure 3C). Additionally, while leptin levels were significantly decreased in PLAT from alcohol-fed animals compared to control animals (*p* = 0.0066) (Figure 3E), there was an increase in leptin levels in SFAT of alcohol-fed animals compared to pair-fed controls (*p* = 0.0496) (Figure 3F). However, circulating levels of leptin in blood and lymph were not significantly different between groups (*p* = 0.5476; *p* = 0.1508) (Figure 3G,H).

### 2.4. Chronic Alcohol Feeding Alters Expression of Genes Involved in PLAT Lipid Metabolism

To evaluate pathways of lipid metabolism in PLAT, we assessed gene expression of adipose triglyceride lipase (ATGL), a main lipase involved in adipose tissue lipolysis, and fatty acid synthase (FAS), a key enzyme for de novo lipogenesis in adipose tissue. We found that adipose triglyceride lipase (ATGL) was modestly elevated in PLAT from alcohol-fed and pair-fed animals, but this difference in expression failed to reach statistical significance (*p* = 0.1347) (Figure 4A). Gene expression of FAS was significantly decreased in PLAT from alcohol-fed animals compared to pair-fed (*p* = 0.0017) (Figure 4B).

### 2.5. Mesenteric Lymph from Chronic Alcohol-Fed Animals Induces IL-6 Expression in Naïve PLAT Explants

To assess the direct impact of lymph contents on adipose tissue, we stimulated naïve PLAT explants with mesenteric lymph from alcohol-fed or pair-fed animals. Measures were normalized to naïve (no-lymph stimulated) control explant. Stimulation with lymph from alcohol-fed or pair-fed animals did not alter adiponectin (*p* = 0.6905) (Figure 5A) or leptin (*p* > 0.999) (Figure 5B) secretion by naïve PLAT explants. However, stimulation with lymph from alcohol-fed animals produced a significant increase in IL-6 expression in naïve PLAT explants (*p* = 0.0286) (Figure 5C).

## 3. Discussion

This study sought to further describe the role of lymph-adipose crosstalk in alcohol-mediated adipose tissue pathophysiology. Our data show that chronic alcohol-fed rats presented PLAT-specific alterations in adiponectin and leptin levels, alterations in the expression of genes involved in lipid metabolic pathways, and decreased lymph adiponectin levels. This was the first study, to our knowledge, to measure adipokine levels and alcohol concentration in mesenteric lymph samples from alcohol-fed rats. These findings were coupled with changes in whole-body glucose homeostasis after alcohol feeding. Using a novel lymph-stimulated naïve PLAT explant design to model lymph leakage, we found that direct stimulation with lymph contents from alcohol-fed animals increased proinflammatory cytokine expression in naïve PLAT, demonstrating the ability of lymph contents to differentially impact naïve adipose tissue.

While most studies investigating the effects of alcohol on adipose tissue focus on the largest and most readily accessible intra-abdominal fat pads, gonadal fat, or SFAT [22,23], our study characterized these effects in mesenteric adipose tissue, or PLAT, and further supported the novel concept of lymph-adipose crosstalk in alcohol-induced pathophysiology. PLAT is thought to be most analogous to human visceral adipose tissue both in its location and physiology, because of its drainage to the portal vein. Our study found that chronic alcohol feeding altered adipokine secretion in a depot-specific manner. Alcohol-fed animals had lower expressions of adiponectin and leptin than control animals in PLAT but not in SFAT. Dysregulated secretion of adipokines in adipose tissue is often linked to inflammation, leading to significant metabolic complications [24]. Bioactive adipokines have both local, within adipose tissue, and distant effects, as these molecules circulate through the portal system to the liver where they modulate metabolic function. Whether these adipokines also circulate through the lymph to exert their effects on distant organs is still to be determined.

We found that chronic alcohol feeding induces alterations in the gene expression of key enzymes in lipogenesis (FAS) and lipolysis (ATGL) in PLAT. Our study showed a significant decrease in FAS and a modest increase in ATGL suggesting a shift toward lipolysis after chronic alcohol feeding. This is supported by published studies showing decreased adipocyte diameter in mesenteric adipose tissue and mesenteric fat pad weight after 8 weeks of alcohol feeding [25]. Because of the venous drainage of PLAT through the portal circulation, FFAs released by PLAT are delivered directly to the liver and can promote ectopic fat accumulation in the liver, interfering with regular organ function [26]. The role of the adipose–liver axis has been described as contributing to alcohol-induced liver pathology such as alcohol-associated liver disease [27]; however, subsequent studies are needed to strengthen the understanding of the role of lymph leakage in this dynamic process.

Interestingly, we did find differences between plasma and lymph levels of adiponectin. Chronic alcohol feeding decreased lymph adiponectin levels while these levels were unchanged in plasma. Both visceral and subcutaneous adipose depots secrete adipokines that contribute to circulating levels [15]. Despite visceral adipose tissue representing a small percentage of total body fat [28] and potentially contributing less to total circulating adipokine levels, adipokine expression in visceral adipose tissue is significantly associated with liver disease, while adipokine expression in SFAT is not [29] reinforcing the importance of PLAT-secreted adipokines. PLAT is largely drained by mesenteric vessels, lymphatic vessels and portal veins, directly exposing the liver to adipokines and other mediators secreted by PLAT [30]. Recent studies have highlighted the mesenteric lymphatic system as a second route of transport for molecules and metabolites from the mesentery [31]; however, subsequent studies are required to better understand its role in alcohol-mediated pathophysiology. Previous studies from our laboratory have found a transient decrease in circulating adiponectin at 30 min post-binge alcohol [19]. It is possible that this alcohol-induced decrease in adiponectin is an acute effect, and a binge dose of alcohol in addition to chronic alcohol feeding may be required to elicit a decrease in circulating adiponectin.

Our previous studies showed impairment in local PLAT insulin-stimulated glucose uptake with alcohol [20]. It has been hypothesized that insulin resistance and dysregulated systemic glucose homeostasis begin in adipose tissue [5]. In this study, we examined the impact of chronic alcohol on whole-body blood glucose homeostasis in response to a glucose load. Our data showed that chronic alcohol feeding impairs glucose tolerance indicating disruption in whole-body glucose homeostasis. Similarly, previously reported hyperinsulinemic-euglycemic clamp studies revealed that chronic four-week alcohol feeding to rats decreased whole-body glucose utilization [22]. Additionally, a recent study also showed impaired glucose tolerance with OGTT after four weeks of alcohol feeding to rats [32]. Our previous findings have shown that alcohol feeding decreases insulin-signaling cascade activation in PLAT but not SFAT [7] and decreases insulin-stimulated glucose uptake in PLAT [20], suggesting a local PLAT insulin resistance that was not apparent in the SFAT. There is evidence that changes in systemic glucose homeostasis and subsequent IR are initiated in insulin-sensitive tissues such as adipose tissue [33]. PLAT insulin-mediated changes have a great impact on systemic glucose utilization because of the capacity of PLAT, a visceral adipose tissue, to modulate systemic metabolism through endocrine and paracrine actions of adipokines and cytokines, as well as through increased flux of FFAs to peripheral tissues such as the liver.

To further dissect the direct effects of lymph on PLAT, we examined the role of lymph-adipose crosstalk in alcohol-mediated adipose tissue dysfunction using a novel naïve PLAT explant model. We found that adipokine levels were unchanged between naïve PLAT stimulated with lymph from alcohol animals compared to lymph from control animals, and neither lymph-stimulated group was different from the non-lymph stimulated control. We found increased levels of IL-6 in naïve PLAT explants stimulated with lymph from alcohol-fed animals compared to control animals. This finding agrees with our previously published work showing a depot-specific increase in IL-6 expression in PLAT in our rodent model of alcohol-induced mesenteric lymphatic leakage [7]. According to multiple reports in the literature, inflammation is linked to and may precede metabolic dysfunction in adipose tissue [34,35,36]. It is possible that our experiment has captured the inflammatory response induced by lymph from alcohol-fed animals. However, longer incubation times may be required to see metabolic changes in the naïve tissue ex vivo. A limitation of this ex vivo design is the duration of explant viability in culture [15,37]. To our knowledge, this is the first study in which whole mesenteric lymph was incubated with tissue explants; others have utilized 3T3-L1 adipocytes for lymph co-culture studies. The first study showed that lymph co-cultured with adipocytes promotes adipogenesis and lipid droplet formation [38]. The second study, similar to ours, showed that lymph from HFD-fed mice co-cultured with differentiated 3T3-L1 adipocytes increased lipid accumulation and insulin resistance, compared to adipocytes co-cultured with lymph from pair-fed mice [21], indicating that lymph contents can differentially impact adipocyte physiology based on treatment conditions. These studies have made the importance of investigating circulating mediators in lymph apparent.

Our study is not without limitations. We are limited by sample size in lymph analyses due to the technical difficulty in obtaining samples. Further, while our naïve lymph-stimulated PLAT explant model is novel, the limited duration of culture necessary to ensure tissue viability may restrict the ability to fully observe metabolic changes induced by lymph-adipose crosstalk. However, our findings that lymph from alcohol-fed animals increased IL-6 expression in naïve PLAT, while lymph from pair-fed animals did not, supporting the concept that inflammatory mediators are present in lymph from alcohol-fed animals and warrant further investigation.

This study has shown that chronic alcohol feeding leads to depot-specific alterations in metabolic profile, reflected by altered adipokine expression and lipid metabolism in PLAT but not SFAT, and impaired systemic glucose tolerance, further supporting our hypothesis of alcohol-induced lymph-adipose crosstalk. Upon co-culture of naïve PLAT explants with lymph from alcohol-fed or lymph from control animals, alcohol-fed animal lymph stimulation led to increased IL-6 levels in naïve PLAT explants, suggesting that lymph from alcohol-fed animals promotes tissue inflammation. Future studies are required to investigate mediators, such as cytokines and gut-derived microbial products, present in lymph from alcohol-fed animals that may be leading to PLAT immunometabolic dysregulation in the setting of mesenteric lymphatic leakage. Moreover, investigating the dynamic cellular interactions between adipocytes and immune cells in PLAT with alcohol feeding could allow us to further elucidate mechanisms of alcohol-induced adipose tissue dysfunction.

## 4. Materials and Methods

### 4.1. Animals and Diet

Male Fisher 344 rats (250–300 g body weight) were housed in a controlled temperature (22 °C) and controlled illumination (12:12 h light–dark cycle) environment. The animals received a Lieber–DeCarli liquid diet (BioServ, Flemington, NJ, USA) with either 36% of caloric intake from alcohol (alcohol-fed) or they were pair-fed (control) with a liquid diet where maltose-dextran was isocalorically substituted for alcohol as previously described [20]. Multiple cohorts were run according to the protocol for experiments. Naïve animals were chow-fed for 10 weeks. Animals were weighed weekly. Rats gained weight over time on both alcohol-containing and control diets. After 10 weeks of feeding, there were no differences in weight change or end weight between groups. Animal studies were approved by the Institutional Animal Care and Use Committee at the Louisiana State University Health Sciences Center (#4544, 8/24/2022, Approved by LSUHSC’s Institutional Animal Care and Use Committee) and were performed in accordance with the guidelines of the NIH Guide for the Care and Use of Laboratory Animals [39].

### 4.2. Intraperitoneal Glucose Tolerance Test

One week prior to sacrifice, alcohol- (n = 6) and pair-fed (n = 4) animals were fasted for 16 h prior to intraperitoneal (IP) injections of 20% dextrose solution at a dose of 2 g/kg. Blood samples (100 microliters) were collected from the tail prior to and at 5, 10 15, 30, 45, 60, 90, and 120 min time points following glucose injection. Blood glucose levels were tested at each interval using the AlphaTRAK glucometer (Zoetis, Kalamazoo, MI, USA) [40].

### 4.3. Lymph Fistula Technique for Mesenteric Lymph Collection

On the day of sacrifice, the lymph fistula procedure was performed as previously described [41,42]. Briefly, animals were anesthetized (ketamine and xylazine (90 and 9 mg/kg, respectively), an abdominal incision was made, and the superior mesenteric lymph duct was cannulated with soft vinyl tubing (medical grade; 0.8 mm OD and 0.5 mm ID; Dural Plastics and Engineering, Dural, Australia). The lymphatic cannula was secured using a drop of methyl cyanoacrylate glue and exteriorized through the right flank. The abdomen was closed with surgical staples, and the animal was placed on a heating pad to maintain body temperature. Mesenteric lymph was allowed to passively flow through the cannula into a 1.5 mL Eppendorf tube on ice for 1 to 2 h; between 500 uL and 1 mL of lymph was collected from each animal. Before sacrifice, the surgical staples were removed to reopen the abdomen and 3 mL of blood was collected from the inferior vena cava and centrifuged at 750× *g* for 15 min for plasma separation. The mesentery was then resected and pinned in a dissection chamber containing 4 °C saline for mesenteric fat (PLAT) isolation. PLAT was carefully excised from the intestine. Inguinal subcutaneous fat was also collected from the left inguinal region. Lymph, plasma, and tissues collected were immediately frozen in liquid nitrogen and stored at −80 °C for downstream analysis. Due to the success rate of lymph collection (70%), only a subset of animals in which lymph collection was successful were used in lymph studies.

### 4.4. Blood/Lymph Ethanol Content Measurements

Frozen 10 uL lymph aliquots were thawed on ice and centrifuged at 500× *g* for 5 min. From all cohorts, lymph and plasma alcohol concentrations were determined in duplicate using the Analox micro-stat GM7 (Analox Inst. Ltd.; Lunenburg, MA, USA).

### 4.5. Adipokine Measurements

PLAT (100 mg of tissue) and SFAT (100 mg of tissue) samples were homogenized in Tissue Protein Extraction Reagent buffer containing protease and phosphatase inhibitors (ThermoScientific, Waltham, MA, USA). Homogenate concentrations of adipokines in alcohol-fed and pair-fed animals were measured using commercially available Rat Adiponectin ELISA (RayBiotech, Peachtree Corners, GA, USA) and Rat Leptin ELISA (RayBiotech, Peachtree Corners, GA, USA) kits. Results were normalized to total protein concentration determined with BCA assay. Because the RayBiotech Adiponectin ELISA was incompatible with plasma samples according to the manufacturer’s protocol, a commercially available Rat Adiponectin ELISA (ALPCO, Salem, NH, USA) was used to measure adiponectin levels in plasma, lymph, and explant culture supernatant samples according to manufacturer’s instructions.

### 4.6. Total RNA Isolation and Real-Time Quantitative PCR (qPCR)

Total RNA was extracted from PLAT using the RNeasy Mini Kit (Qiagen, Germantown, MD, USA), as per the manufacturer’s instructions. cDNA was synthesized from 0.1 μg of the resulting total RNA using the Maxima First Strand cDNA Synthesis Kit (ThermoScientific, Waltham, MA, USA), in accordance with the manufacturer’s instructions. Primers were designed to span exon–exon junctions, purchased from Integrated DNA Technologies (IDT, Coralville, IA, USA; Table 1), and used at a concentration of 500 nmol. The final reactions were made to a total volume of 20 μL with Quantitect SyBr Green PCR kit and DNase RNase-free water (Qiagen, Germantown, MD, USA). All reactions were carried out in duplicate on a CFX96 system (Bio-Rad Laboratories, Hercules, CA, USA) for qPCR detection. qPCR data were analyzed using the comparative Ct (delta–delta-Ct, DDCT) method. Target genes were compared with the endogenous control, ribosomal protein S13 (RPS13), and alcohol values normalized to control values [43].

### 4.7. Perilymphatic Adipose Tissue Explant Culture

The mesenteric lymph from alcohol (n = 5) and control (n = 5) animals were collected using the lymph fistula technique described above. PLAT explants were harvested from naïve, age-matched, chow-fed animals, split into three explants, and stimulated with lymph (5% *v/v* in 1 mL of DMEM) from alcohol-fed animals, pair-fed animals, or no lymph to serve as a vehicle. After 48 h, PLAT explants were collected, frozen, and stored at −80 °C for analysis.

### 4.8. Statistical Data Analysis

Data are summarized as means ± standard error of the mean (SEM). An unpaired *t*-test was utilized to identify significant differences between data from alcohol- and pair-fed groups. Differences in glucose tolerance between alcohol- and pair-fed groups were measured using two-way ANOVA. The association between blood alcohol concentration and lymph alcohol concentrations was determined using Pearson’s correlation coefficient. All statistical analyses were performed using GraphPad Prism 9.0 software, with significance set at *p* < 0.05.

## Figures and Tables

**Figure 1 ijms-25-10811-f001:**
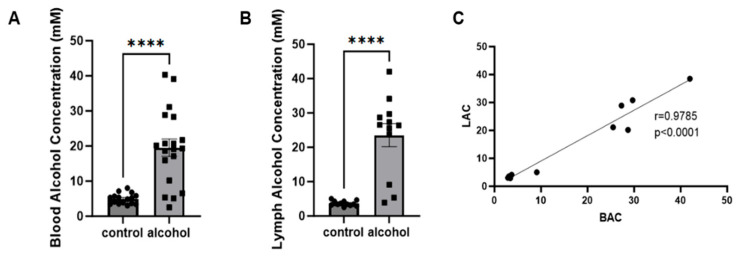
Alcohol concentrations were measured in plasma (**A**) and mesenteric lymph (**B**) samples using the Analox. Student’s *t*-test was used to assess statistical differences between alcohol- and pair-fed groups. Pearson’s correlation coefficient was used to assess the association between blood alcohol concentration (BAC) and lymph alcohol concentration (LAC) (**C**) **** *p* < 0.0001.

**Figure 2 ijms-25-10811-f002:**
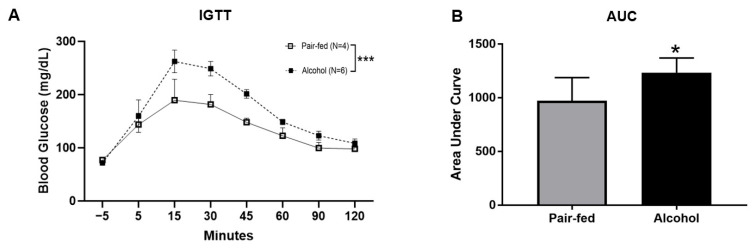
Blood glucose levels over time (min) in alcohol-fed and pair-fed rats prior to and following an IP injection of glucose (**A**) Total area under the curve from IGTT (**B**) Two-way ANOVA and Student’s *t*-test were used to assess statistical differences between groups. *** *p* < 0.005; * *p* < 0.05.

**Figure 3 ijms-25-10811-f003:**
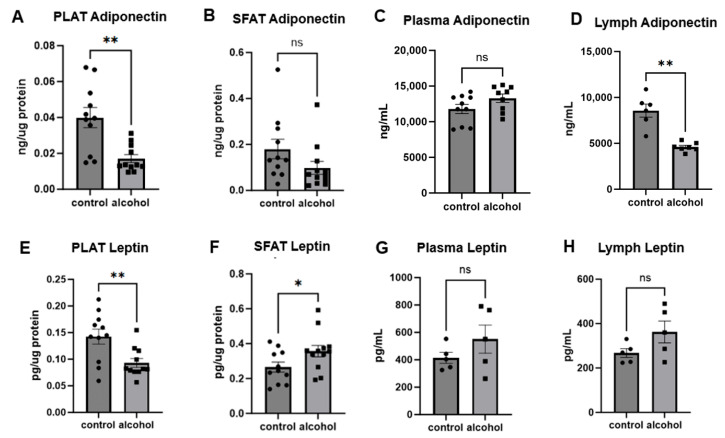
Adiponectin was measured in PLAT (**A**) and SFAT (**B**), plasma (**C**), and lymph (**D**) and compared between alcohol- and pair-fed groups. Leptin was measured in PLAT (**E**), SFAT (**F**), plasma (**G**), and lymph (**H**) and compared between alcohol- and pair-fed groups. Student’s *t*-test was used to assess statistical differences between groups. ** *p* < 0.005; * *p* < 0.05; ns = not significant.

**Figure 4 ijms-25-10811-f004:**
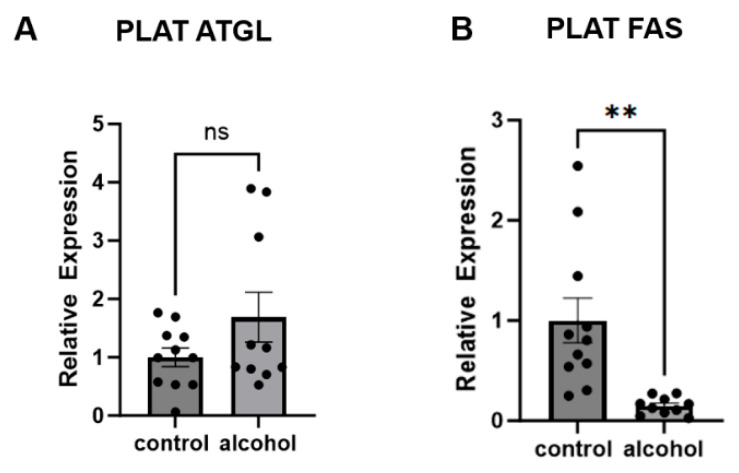
Gene expression of PLAT ATGL (**A**) and FAS (**B**) relative to RPS13 in control and alcohol-fed animals. Student’s *t*-test was used to assess statistical differences between groups. ** *p* < 0.01; ns = not significant.

**Figure 5 ijms-25-10811-f005:**
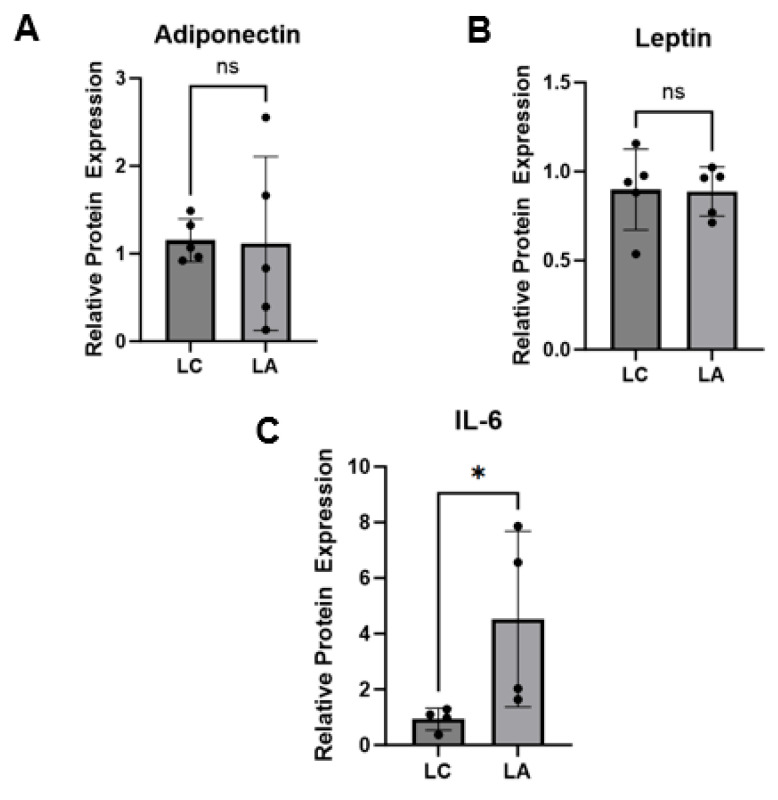
Protein expression of adiponectin (**A**), leptin (**B**), and IL-6 (**C**) was compared between naïve PLAT explants stimulated with lymph from control animals (LC) and PLAT stimulated with lymph from alcohol-treated animals (LA) relative to naïve (no-lymph stimulated) control explant. Mann–Whitney test was used to assess statistical differences between groups. * *p* < 0.05; ns = not significant.

**Table 1 ijms-25-10811-t001:** Primer sets used in RT-PCR analysis.

Gene	Forward Primer	Reverse Primer
Fatty Acid Synthase (*fasn*)	GACCCTGACTCCAAGTTATTC	GCAGCTCCTTGTATACTTCTC
Adipose Triglyceride Lipase (*pnpla2*)	GTACCCTATACTCTGCCACT	TACCTGTCTGCTCCTTCAT
RPS13	GCACCTTGAGAGGAACAGAA	GAGCACCCGCTTAGTCTTATAG

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
