# Peer review of "The Role of Lymph-Adipose Crosstalk in Alcohol-Induced Perilymphatic Adipose Tissue Dysfunction"

_ijms, 2024, doi:10.3390/ijms251910811_

Round 1

Reviewer 1 Report

Comments and Suggestions for Authors ======== REVIEW REPORT ===============  

In this manuscript by Weaver et al., the authors have further explored the lymph-adipose crosstalk in the context of alcohol-induced PLAT dysfunction. This manuscript is presented by an expert group in this topic. The manuscript is well-written and easy to follow. This study further advances our knowledge of what is known about alcohol-induced lymph-adipose dysfunction and for that, I commend the authors for their continuous efforts in this topic of high clinical significance; although it is also highly descriptive, and the functional/mechanistic components could be improved. Here are some comments, in no particular order, that may help the authors improve their submission:

1) Throughout the figures, the authors sometimes use asterisks to indicate significance, while in other cases, the actual p-values are indicated. For consistency, please use one or the other, I would recommend including actual p-values. 

2) I would be curious about what the p-value for Figure 3B was. The trend of the data presented seems rather strong, and while the data was found not significantly different, it appears that results and conclusions made based on these results are relative to the sample size. Did the authors perform power tests to determine minimum sample size in these experiments? 

3) Similarly, given the variability of the data, increasing sample size for the assessment of plasma leptin and lymph leptin levels in Figure 3G and 3G may help better support the conclusions. These changes in leptin and their functional impact may be underestimated here. 

4) The format of Figure 5 is slightly different than the rest of the figures, please make sure all your figures are consistent throughout the manuscript.

5) The changes in PLAT ATGL and FAS are very interesting, it would significantly improved the study if the authors validated these findings via protein expression and function. And/or if the authors explored further mechanistic impacts related to these gene expression changes. 

6) Have the authors considered ways to prevent and/or revert the lymph-adipose dysfunction induced by alcohol? Alternatively, have the authors considered testing the effects of alcohol-induced lymph-adipose dysfunction in animal models known to display hyperpermeability? For instance, rat models for metabolic syndrome, diet-induced obese rodents, Prox1+/-, db/db, or ApoE-KO mice, among others, which display lymphatic hyperpermeability, do the authors anticipate that these mice would be more susceptible to the effects of alcohol-induced dysfunction? And by extrapolation, do the authors think that, with the global obesity epidemic and the increase in alcohol consumption world-wide, obesity and alcohol-induced mechanisms of dysfunction would synergize to cause further alterations to the lymphatic and adipose systems? 

  ====================================  

Reviewer 2 Report

Comments and Suggestions for Authors

Dear Authors,

I recommend the publication of this manuscript, as it provides valuable information regarding the mechanisms of alcohol-induced metabolic dysfunction, particularly at the level of perilymphatic adipose tissue (PLAT).

The study's findings contribute significantly to the current understanding of alcohol-related metabolic disorders by focusing on how chronic alcohol consumption leads to lymphatic drainage, adipose tissue inflammation, and systemic glucose dysregulation. The innovative ex vivo model used to explore lymph-induced inflammation in PLAT adds novelty to the field, and the data present a compelling case for further research into the specific lymph components involved.

The study is well structured, the arguments presented are clear and the discussions and conclusions are relevant.

I believe this paper will be of great interest to researchers focused on metabolic health and related fields.

Reviewer 3 Report

Comments and Suggestions for Authors

This study explores alcohol-induced lymphatic leakage and its role in perilymphatic adipose tissue (PLAT) dysfunction. It demonstrates that chronic alcohol consumption alters adipokine levels, induces inflammation, and impairs glucose tolerance in rats, highlighting lymph-adipose crosstalk as a key factor in alcohol-related metabolic disturbances. Overall, the paper offers a solid foundation for further research on alcohol-induced adipose tissue dysfunction and inflammation. However, there are a few areas where the scientific rigor could be strengthened. My comments are as follows:

 1.     The authors should investigate insulin resistance markers. While the study shows impaired glucose tolerance and decreased adiponectin levels, markers directly related to insulin resistance, such as insulin receptor substrate (IRS), glucose transporter-4 (GLUT-4), and phosphorylated AKT, were not assessed. Including these markers could provide clearer insights into how alcohol-induced lymphatic leakage affects insulin signaling pathways in PLAT and whether this dysfunction is directly contributing to systemic insulin resistance.

2.     The leptin/adiponectin ratio often correlates with metabolic disorders in humans, but in this study, that association isn't evident. While the authors report reduced leptin and adiponectin in PLAT, they did not explore the leptin/adiponectin ratio in relation to systemic insulin resistance or metabolic outcomes. The authors should not only focus on depot-specific effects of alcohol on adipokines but also their systemic roles, or species-specific differences between humans and rats in this context.

3.     The study examines changes after 10 weeks of alcohol feeding but does not address long-term consequences or potential reversibility of the observed adipose dysfunction following alcohol cessation.

4. The viability of PLAT explants in culture might be constrained by the 48-hour incubation period used in the ex vivo experiments. This limited duration could restrict the ability to fully observe the metabolic changes induced by lymph-adipose crosstalk. A longer incubation time may be required to capture more pronounced metabolic alterations in PLAT explants.

Comments on the Quality of English Language

The quality of the English language in the paper is generally good but could benefit from improvements in clarity, grammar, and sentence structure in some areas. For example, certain sentences are lengthy and could be made more concise, while others could use smoother transitions or rephrasing for better readability. Enhancing consistency in tenses and simplifying complex phrases could also improve overall readability. Additionally, addressing minor issues in punctuation would elevate the quality of the writing.

Round 2

Reviewer 3 Report

Comments and Suggestions for Authors

The authors answered my comments point by point and improved the quality of the manuscript. It can be accepted for publication.